# Medical education in times of COVID-19: German students' expectations – A cross-sectional study

Teresa Loda[1], Tobias Löffler[2], Rebecca Erschens[1], Stephan Zipfel[1,2], Anne Herrmann-Werner[1,3]*

**1** Department of Internal Medicine VI/Psychosomatic Medicine and Psychotherapy, University Hospital Tuebingen, Tuebingen, Germany, **2** Faculty of Medicine, Eberhard-Karls University of Tuebingen, Tuebingen, Germany, **3** Competence Center for University Teaching in Medicine, Faculty of Medicine, University of Tuebingen, Tuebingen, Germany

* anne.herrmann-werner@med.uni-tuebingen.de

**Data Availability Statement:** All relevant data are within the manuscript.

**Funding:** We acknowledge support by Deutsche Forschungsgemeinschaft and Open Access

## Abstract

### Background

Since the COVID-19 pandemic has affected the education of medical students, medical faculties have faced the challenge of adapting instruction to digital platforms. Although medical students are willing to support pandemic response efforts, how the crisis will affect their medical training remains uncertain. Thus, in this study, we investigated the teaching- and learning-related stressors and expectations of medical students in Germany during the COVID-19 pandemic.

### Methods

A cross-sectional survey was distributed online to undergraduate medical students at medical faculties in Germany. Students answered questions about COVID-19 and teaching (on a 7-point Likert scale from 0 ("not at all") to 6 ("completely")) and completed mental well-being measurements, including the State–Trait Anxiety Inventory (STAI), the Generalised Anxiety Disorder scale (GAD-7) and the Perceived Health Questionnaire (PHQ-9). Descriptive data analysis, a $t$-test and Pearson correlations were performed to process the data.

### Results

Medical students felt well-informed about COVID-19 in general ($M = 5.64$, $SD = 1.28$) and in the medical context ($M = 5.14$, $SD = 1.34$) but significantly less informed about the pandemic in the academic context, $M = 2.47$, $SD = 1.49$, $t(371) = 31.98$, $p < .001$. Their distress levels were high (STAI: $M = 45.12$, $SD = 4.73$) and significantly correlated with the academic context ($r_p = .164$, $p < .01$) but not their private lives. Concerning how they were taught, they most often expected online lectures (91.7%) and live broadcasts (67.2%) and less often expected innovative digital teaching strategies, including serious games (17.3%) and virtual-reality exercises (16.7%).

Publishing Fund of University of Tuebingen. The authors did not receive any salary from the funders. The funders had no role in study design, data collection and analysis, decision to publish, or preparation of the manuscript.

**Competing interests:** The authors have declared that no competing interests exist.

## Discussion

Medical students seem to be aware of the COVID-19 pandemic and its consequences for academic and healthcare contexts. They also seem to think that their teachers will enhance their digital competencies during the pandemic. Therefore, faculties of medicine need to rapidly and adequately digitalise their approaches to teaching.

## Introduction

The COVID-19 pandemic has dramatically changed countless aspects of life around the world. The education of medical students is no exception, and in response to the pandemic, faculties of medicine worldwide have faced the challenge of adapting instruction to digital platforms. Beyond that, the role of medical students, as clinicians in training, in possibly helping to manage the pandemic has been discussed, especially regarding whether they should graduate early to serve on the frontlines [1–3]. That possibility is supported by the fact that medical students remain updated about the COVID-19 pandemic and demonstrate high levels of related knowledge and preventive behaviours [4]. Indeed, medical students have even developed student-led organisations to efficiently mobilise their peers who are interested in participating in COVID-19 response efforts [5–7]. In Germany in particular, within days of the government's request for assistance, 20,000 German medical students agreed to offer their support in combatting the pandemic [8].

Despite that overwhelming solidarity, medical students, as students of all disciplines, have had to grapple with the hallmark feature of the COVID-19 pandemic: uncertainty due to novelty of the virus. At present, no one can reliably predict how the pandemic will unfold or what rapid, flexible, ad hoc decision-making will be necessary, and such uncertainty affects medical training at all levels [9]. Classroom lectures, seminars and clinical placements can no longer be performed in the usual face-to-face settings, and some observers have even called for care activities directly provided to patients, usually a core part of medical training, to be banned [10, 11]. At the same time, the debate continues over whether final-year medical students should be fast-tracked through their studies so that they can support the healthcare system or even begin practice before being licensed [12, 13].

To maintain the quality of medical training under the circumstances of the COVID-19 pandemic, medical educators need to think outside the box [9]. As one set of solutions, digital technologies have been used to support innovative teaching on e-learning platforms, virtual training or video-conferencing [9, 14–16]. Several authors have also suggested combining technology-enhanced learning experiences with traditional ones [9, 14, 15, 17]. Nevertheless, little is known about medical students' perspectives on continuing their training during the summer semester 2020 as the pandemic persists.

### Aim of the study

In the study reported here, we aimed to investigate the teaching- and learning-related stressors and expectations of medical students in Germany during the COVID-19 pandemic. We also assessed their current distress levels and strategies of coping with such stressful situations. To the best of our knowledge, our study was the first to examine those topics.

## Methods

### Study design, participants and procedure

The explanatory, cross-sectional study was performed across faculties of medicine in Germany. All faculty-affiliated undergraduate medical students were invited to participate in the online survey, which commenced in late March 2020 and lasted three weeks.

### Ethics

The study received ethics approval from the Ethics Committee of Tuebingen Medical Faculty (no. 314/2020BO2). Participation was voluntary, and students did not receive any reimbursement for participating. All participants provided their written informed consent, and all of their responses and data were kept anonymous.

### Measurements

**Demographic characteristics.** Demographic data (i.e. gender, age, year of study and place of study) were collected via the survey.

**Topics concerning COVID-19.** Students rated their level of knowledge about COVID-19 in general, as well as in the medical and academic contexts, on a 7-point Likert scale ranging from 0 ("not at all") to 6 ("completely"). They also answered multiple-option questions about how they had accessed that information. They rated their fear and felt probability of being infected with COVID-19 (0% to 100%); their ability to handle a crisis in general; the pandemic's general and specific burdens on their private and academic lives; their distress in response to the pandemic; and their access to means of coping with the pandemic. Last, they indicated whether they felt ready to provide support during the crisis (i.e. "Yes" or "No" question) and, if so, then in what ways (i.e. multiple-option question).

**Mental well-being measurements.** For mental well-being measurements of the students' personal experiences during the COVID-19 pandemic, three validated questionnaires were used:

1. The State–Trait Anxiety Inventory (STAI) [18], particularly the "State anxiety" dimension, to measure the intensity to which they felt distress on a 4-point scale from 1 ("not at all") to 4 ("very much so");

2. The Generalised Anxiety Disorder (GAD7) [19] and Perceived Health Questionnaire (PHQ9) [20] to measure anxious and depressive symptoms on a 4-point Likert scale from 0 ("not at all") to 3 ("nearly every day") [19–21]; and

3. The Internal External Locus of Control scale [22] to measure perceived control over the environment on a 5-point Likert scale from 0 ("not at all") to 5 ("completely").

**Education.** Students rated their desired and expected course content for the summer semester 2020 as well as possible long-term changes in their medical education due to the circumstances of the pandemic (i.e. multiple-option questions). On a 7-point Likert scale from 0 ("not at all") to 6 ("completely"), they rated the consideration of their study-related needs by federal student representatives and their worry over losing a semester. Added to that, they reported whether their state examinations had been postponed due to the pandemic (i.e. "Yes" or "No" question) and their expectations of the national government, their faculties of medicine and their teachers (i.e. multiple-option questions and open comments). Last, they stated stressors relevant to studying during the pandemic (i.e. open comments).

All items concerning COVID-19 and education were based on literature [1, 4, 5, 23].

## Data analysis

The normal distribution of the data was confirmed by using the Kolmogorov–Smirnov test. Descriptive data, including mean values (*M*), standard deviations (*SD*), sum scores, frequencies and percentages of relevant factors, were calculated, and any missing value was replaced with the mean value. To compare the results, a *t* test for independent samples and Pearson correlations were performed. The Statistical Package for the Social Sciences version 26.0 (IBM, Armonk, NY, USA) was used for data analysis. The level of significance was set to $p < .05$. All questionnaires that were filled in at least 80% were included in the study.

## Results

### Demographic characteristics

Of the 679 students who received the survey, 372 responded in full, for a response rate of 55.1%. By gender, 279 were women (75.0%), 92 were men (24.7%), and one was other (0.3%), and by age, they were 23.92 years old on average (*SD* = 4.21, range = 18–48). Students came from all levels of medical school: first year of study (*n* = 57, 15.3%), second year (*n* = 79, 21.2%), third year (*n* = 64, 17.2%), fourth year (*n* = 58, 15.6%), fifth year (*n* = 78, 21.0%) and final year (*n* = 36, 9.7%). Of 38 German faculties of medicine, 28 were represented.

### Topics concerning COVID-19

**Level of knowledge in general, medical and academic contexts.** Students felt well-informed about COVID-19 in general (*M* = 5.64, *SD* = 1.28) and in the medical context (*M* = 5.14, *SD* = 1.34). However, they felt significantly less informed about the virus and pandemic in their academic contexts, *M* = 2.47, *SD* = 1.49, $t(371) = 31.98$, $p < .001$. Students reported retrieving their information mostly from governmental webpages (*n* = 248, 66.7%) and television (*n* = 238, 64.0%).

Students rated their felt probability of getting infected with COVID-19 at around 58% and their fear of being infected at 29.5%, both on average. Most students (*n* = 270, 72.6%) agreed that they were able to cope with the crisis (*M* = 5.06, *SD* = 1.12), and fewer than half (*n* = 139, 37.4%) reported feeling distressed in their private lives due to the pandemic (*M* = 3.95, *SD* = 1.52). By contrast, the majority of students (*n* = 229, 61.6%) reported feeling significantly more distressed over their studies due to COVID-19, *M* = 4.82, *SD* = 1.71, $t(371) = 54.31$, $p < .001$. Less than a third (*n* = 116, 31.2%) wanted recommendations on how to cope with their distress (*M* = 3.47, *SD* = 1.94). Slightly more students, $t(371) = 30.93$, $p < .001$, preferred recommendations about relaxation techniques (*n* = 125, 33.6%, *M* = 3.33, *SD* = 1.97) or sought psychotherapeutic interviews (*n* = 69, 18.5%, *M* = 2.83, *SD* = 1.77). Last, 89.8% of students were willing to offer support during the COVID-19 pandemic; 271 (72.8%) were willing to treat patients without COVID-19, and 220 (59.1%) were willing to treat patients with the virus (Fig 1).

**Psychometric characteristics.** The current distress level among students was high (cut-off > 43) in view of STAI scores (*M* = 45.12, *SD* = 4.73). They also showed mild (cut-off > 5) anxious and depressive symptoms ($M_{GAD7} = 5.73$, $SD_{GAD7} = 4.70$; $M_{PHQ9} = 5.59$, $SD_{PHQ9} = 4.66$). Regarding their locus of control, the students on average presented a high (cut-off > 7) internal level of control (*M* = 8.19, *SD* = 1.31) and a normal external level of control (*M* = 4.31, *SD* = 1.48). The mean STAI sum score was significantly correlated with the students' academic context ($r_p = 0.164$, $p < .01$) but not their private lives.

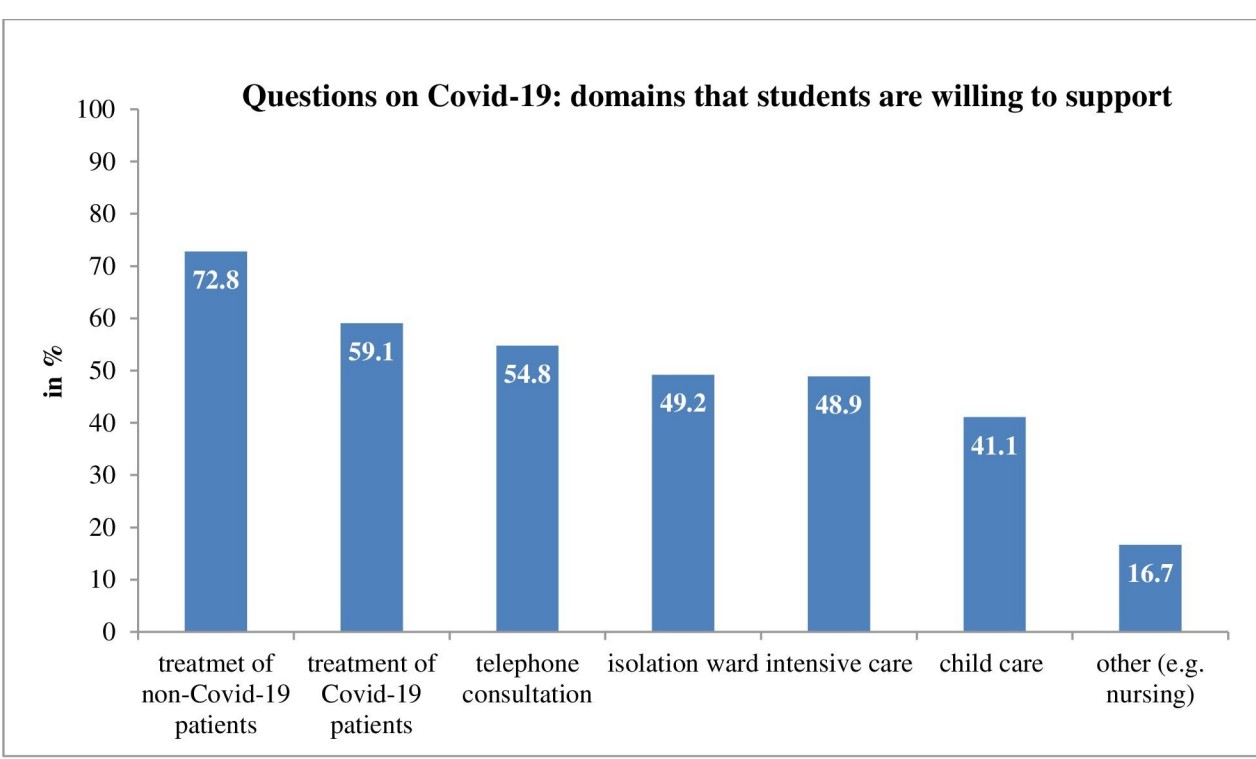

**Fig 1. Medical students' rating of domains they are willing to support during COVID-19 pandemic.**

**Education.** Students rated their desired and expected content for the summer semester 2020, as well as anticipated possible long-term changes in medical education, as shown in Table 1.

Students rated the consideration of their study-related needs significantly higher, $t(368) = 72.46$, $p < .001$, by federal student representatives ($M = 3.66$, $SD = 0.97$) than the government ($M = 2.18$, $SD = 1.04$). They feared missing study materials ($M = 4.01$, $SD = 1.87$) and not having sufficient protective gear when working in healthcare ($M = 4.72$, $SD = 1.68$); however, they were less afraid of missing aspects of their daily lives, $M = 2.42$, $SD = 1.49$, $t(371) = 31.19$, $p < .01$. They were also afraid of losing a semester due to COVID-19 ($M = 4.21$, $SD = 2.18$). Only 40 students (10.8%) reported that their state examinations had been postponed.

In their comments, the students reported general uncertainty and scarcity of information ($n = 173$) as their greatest stressors, followed by more specific study-related worries about their state examinations, practical years and training terms abroad ($n = 54$). They also worried about the disadvantages of studying online, which meant having less social contact, having more complicated interactions with teachers and missing training in clinical skills ($n = 43$).

When working with patients, most students sought content ($n = 320$, 86.0%) and inter-professional ($n = 242$, 65.1%) discussions about patients' cases, and exactly half of them ($n = 188$) wanted more information on resilience training. Furthermore, students expected their teachers to be willing to enhance their digital competencies ($n = 264$, 71%) and to be lenient with them about examinations ($n = 247$, 66.4%), as detailed in Fig 2.

In their comments, the students reported wanting more transparency, clarity and communication ($n = 65$) from the government. They indicated preferring consistent regulations about academic study, schools, examinations, hospitals and social life ($n = 28$). They also expected the government to provide sufficient safety precautions and protective equipment ($n = 27$).

**Table 1. Medical students' ratings of teaching approaches that should be used during the COVID-19 pandemic, will be implemented during the pandemic and will remain after pandemic.**

| Approach | Implementation | *n* | % |
|---|---|---|---|
| **Podcasts** | Should be used during the pandemic | 219 | 58.9 |
| | Will be implemented | 65 | 17.5 |
| | Will remain after the pandemic | 97 | 26.1 |
| **Online lectures** | Should be used during the pandemic | 341 | 91.7 |
| | Will be implemented | 297 | 79.8 |
| | Will remain after the pandemic | 281 | 75.5 |
| **Independent collaborative work** | Should be used during the pandemic | 163 | 43.8 |
| | Will be implemented | 244 | 65.6 |
| | Will remain after the pandemic | 129 | 34.7 |
| **Live broadcasts** | Should be used during the pandemic | 250 | 67.2 |
| | Will be implemented | 119 | 32.0 |
| | Will remain after the pandemic | 108 | 29.0 |
| **Online chats with teachers** | Should be used during the pandemic | 192 | 51.6 |
| | Will be implemented | 89 | 23.9 |
| | Will remain after the pandemic | 66 | 17.7 |
| **Online chats with chatbots** | Should be used during the pandemic | 18 | 4.8 |
| | Will be implemented | 5 | 1.3 |
| | Will remain after the pandemic | 7 | 1.9 |
| **Online exams** | Should be used during the pandemic | 161 | 43.3 |
| | Will be implemented | 40 | 10.8 |
| | Will remain after the pandemic | 65 | 17.5 |
| **Online collaborative work** | Should be used during the pandemic | 119 | 32.0 |
| | Will be implemented | 24 | 6.5 |
| | Will remain after the pandemic | 60 | 16.1 |
| **Virtual- or augmented-reality exercises** | Should be used during the pandemic | 62 | 16.7 |
| | Will be implemented | 1 | 0.3 |
| | Will remain after the pandemic | 21 | 5.6 |
| **Serious games** | Should be used during the pandemic | 64 | 17.3 |
| | Will be implemented | 3 | 0.8 |
| | Will remain after the pandemic | 12 | 3.3 |
| **Asynchronous interactive formats** | Should be used during the pandemic | 125 | 33.9 |
| | Will be implemented | 24 | 6.5 |
| | Will remain after the pandemic | 32 | 8.7 |
| **Other (e.g. recommended literature readings)** | Should be used during the pandemic | 21 | 5.6 |
| | Will be implemented | 39 | 10.5 |
| | Will remain after the pandemic | 33 | 8.9 |

Concerning their faculties of medicine, the students reported wanting more communication, clarity and information about the summer semester 2020 in light of the COVID-19 pandemic, particularly about their examinations (*n* = 131). They were also afraid of the disadvantages that final-year students would face (*n* = 43). They wanted to play an active role in necessary adaptations and demanded thoughtfulness, especially about students with special situations—for example, ones helping out in the crisis or home-schooling their children (*n* = 33).

Finally, students expected their faculties of medicine to provide good online offerings for teaching and practical alternatives for training and examinations (*n* = 23).

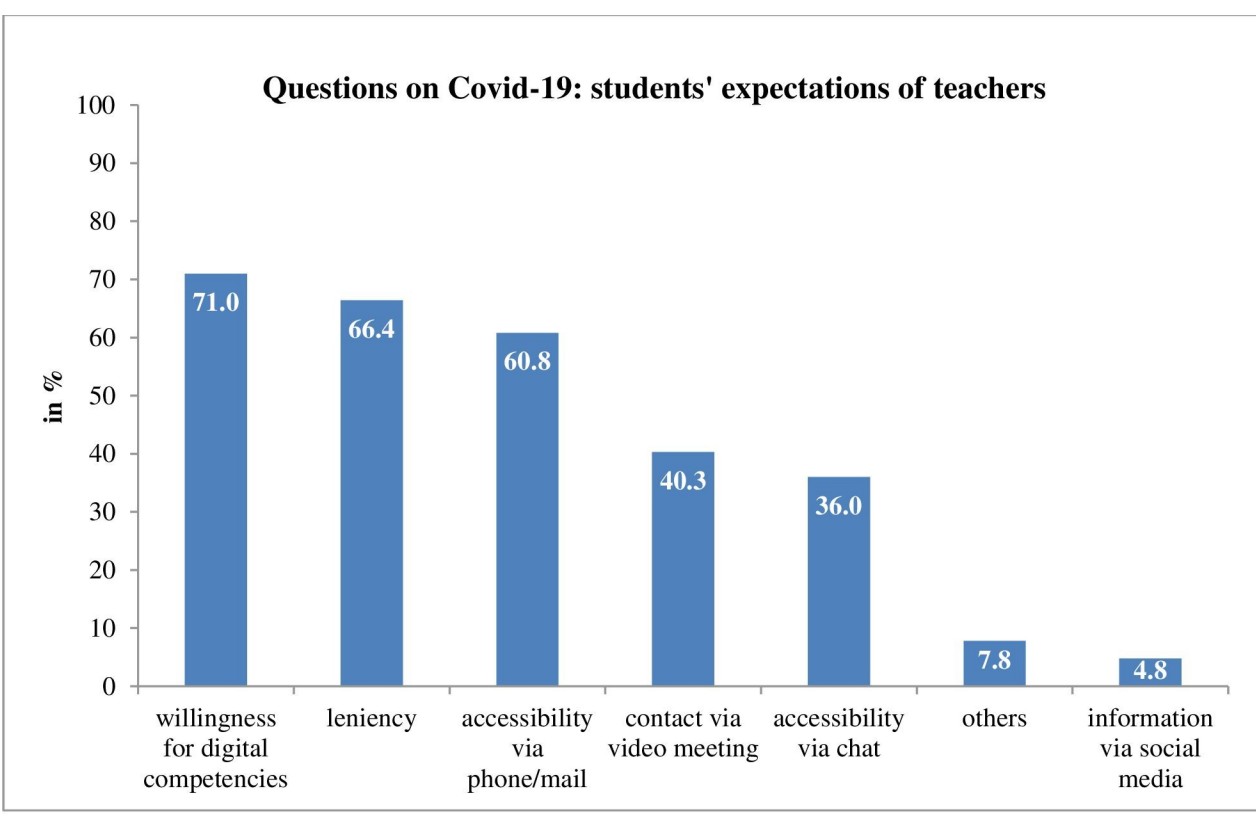

**Fig 2. Medical students' rating of their expectations of teachers.**

## Discussion

This study revealed the teaching- and learning-related expectations and stressors of medical students in Germany regarding the COVID-19 pandemic. The results showed that the students were highly aware of the pandemic and its consequences for medical and academic contexts.

### Medical students' perspective on COVID-19

The results indicated that medical students were well-informed about COVID-19 in general [4]. Most of them (89.8%) were willing to provide support during the pandemic, and 59.1% were even willing to treat patients with COVID-19. Such high willingness amongst medical students to participate in pandemic response efforts has previously been reported by other national and international surveys [5, 8]. At the same time, the study's medical students reported being concerned about their education because they felt less informed about how it would proceed [24]. This seems reasonable as most medical faculties were struggling themselves with this pandemic and its accompanying issues. They also reported that their study-related needs were considered more by federal student representatives than the government, and they indicated wanting more transparency, clarity and communication from the government. Having information or being informed may thus present a key element of being able to cope with the COVID-19 pandemic, for senior medical students and students working in emergency rooms were less afraid of getting infected [4]. Thus, frequent and timely information on any changes made or even merely discussed concerning study-related issues might contribute to lessening students' distress level. It

seems to be secondary who is providing this information: besides established institutions including federal government, universities and faculties, students' representatives may offer a valuable source of information.

Medical students are concerned on high distress level [25, 26]. This study showed that the medical students' current distress level was high compared to a reference group [18]. Such results suggest that high distress is associated with study-related concerns and less to matters in their personal lives due to COVID-19. Most students were also afraid of losing a semester due to the pandemic.

## Education during the COVID-19 pandemic

The COVID-19 pandemic poses two particular challenges for medical education: the necessity to rapidly and adequately digitalise teaching content and the possibility of integrating medical students into the healthcare system [27].

The medical students in the study expected that traditional teaching approaches would be transferred online (e.g. online lectures and live broadcasts) but did not anticipate the use of more innovative teaching tools (e.g. virtual- and augmented-reality exercises and serious games), which reflects the fact that medical training in Germany continues to use conventional modes of instruction. That there was little hope for more innovative, creative digital teaching may be due to experiences thus far in which a mixture of reservations, technical problems and legal requirements hindered instruction.

Nevertheless, the students thought that their teachers were willing to enhance their digital competencies and that some online teaching formats (e.g. online lectures and live broadcasts) would persist after the COVID-19 pandemic [27]. Those findings support the results of Theoret and Ming [8], who found that online teaching and continued online communication may become pillars of medical training [28]. Online teaching has indeed been shown to foster self-learning, be as successful as traditional didactics and provide an enjoyable experience for participants [28–30].

Altogether, although most German medical teachers are directly involved in providing patient care and thus currently experiencing considerable stress, the COVID-19 pandemic offers numerous opportunities like VR technologies [31]. It may even promote long-lasting changes towards sharing educational resources worldwide and funding schemes. It can be hoped that the current wave of digitalisation will not recede but push medical education into a real digital transformation [32].

Simultaneously, medical students worried about the disadvantages of studying online. Thus, students need to be prepared to deal with the necessary digital transformation of medical education. Furthermore, not all parts of medical education should be digitalised. Classic teaching formats like bed-side teaching or clinical examination course should be preserved in their current form.

## Strengths and limitations

This survey strengthens the perspectives of medical students regarding their medical education during COVID-19 including a huge sample of almost 700 medical students and half of the German medical faculties. One limitation of our survey was that the results were not generalizable, because we surveyed only medical students in Germany. In the limitations, we also need to consider a selection bias. Surely, students that were interested in this topic were more willing to complete this questionnaire. Further, 75% of our medical students were female which is not representative (11% higher). Also, although method of contact was the same for each medical school, some medical faculties had no participating students. This may pose a potential

source of bias; however, analyses yielded no explanatory patterns. Furthermore, we did not assess the perspectives of teachers' or faculties of medicine on teaching during the COVID-19 pandemic. Future studies should focus on how teachers and faculties of medicine intend to adapt medical education during the pandemic and beyond, as well as the challenges that they face in the process.

## Conclusion

This study summarised the expectations and stressors of undergraduate medical students during the COVID-19 pandemic. Medical students are well-informed about COVID-19 and willing to support pandemic response efforts, although they may be distressed about their studies. They desire more communication, clarity and information about the summer semester 2020, especially about their examinations. At the same time, they expected that their teachers would enhance their digital competencies in order to adequately adapt instruction. In sum, the COVID-19 pandemic offers numerous opportunities to adapt medical education and promote a lasting digital transformation that allows transparency and communication between medical students and their teachers.

## Acknowledgments

We would like to thank the German national student body (bvmd) for distributing the online link for the survey to local faculty student bodies.

## Author Contributions

**Conceptualization:** Teresa Loda, Anne Herrmann-Werner.

**Formal analysis:** Teresa Loda, Rebecca Erschens.

**Investigation:** Teresa Loda, Tobias Löffler, Rebecca Erschens, Anne Herrmann-Werner.

**Project administration:** Stephan Zipfel, Anne Herrmann-Werner.

**Resources:** Tobias Löffler.

**Supervision:** Stephan Zipfel.

**Writing – original draft:** Teresa Loda, Anne Herrmann-Werner.

**Writing – review & editing:** Tobias Löffler, Rebecca Erschens, Stephan Zipfel.

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
