## [Decision Letter · Decision Letter 0]

4 Sep 2020

PONE-D-20-21365

Medical education in times of Covid-19: German students’ expectations– a cross-sectional study

PLOS ONE

Dear Dr. Herrmann-Werner,

Thank you for submitting your manuscript to PLOS ONE. After careful consideration, we feel that it has merit but does not fully meet PLOS ONE’s publication criteria as it currently stands. Therefore, we invite you to submit a revised version of the manuscript that addresses the points raised during the review process.

We look forward to receiving your revised manuscript.

Kind regards,

Jenny Wilkinson, PhD

Academic Editor

PLOS ONE

Journal Requirements:

3. Thank you for including the following funding information in the acknowledgements section of your manuscript; "We acknowledge support by Deutsche Forschungsgemeinschaft and Open Access Publishing Fund of University of Tuebingen."

"The authors received no specific funding for this work "

4. Please include your tables as part of your main manuscript and remove the individual files. Please note that supplementary tables (should remain/ be uploaded) as separate "supporting information" files

Additional Editor Comments (if provided):

Thank you for your submission, two reviewers have provided comments to improve the work. More description is needed for the questionnaire, particularly methods of validations and how various types of bias that may be present in the work may be controlled. For example, differences in key characteristics between respondents and non-respondents, whether the finishing are skewed due to one or more medical school being over-represented in the respondents and whether consideration was given to the possible impact of social desirability (i.e. respondents proving responses they deemed socially desirable).

Although the reviewers noted that Table 1 was missing I have checked the submission and this was included.

Reviewers' comments:

Reviewer's Responses to Questions

**Comments to the Author**

1. Is the manuscript technically sound, and do the data support the conclusions?

Reviewer #1: Yes

Reviewer #2: Partly

2. Has the statistical analysis been performed appropriately and rigorously? 

Reviewer #1: Yes

Reviewer #2: No

3. Have the authors made all data underlying the findings in their manuscript fully available?

Reviewer #1: Yes

Reviewer #2: No

4. Is the manuscript presented in an intelligible fashion and written in standard English?

Reviewer #1: Yes

Reviewer #2: Yes

5. Review Comments to the Author

Reviewer #1: Note: Table 1 was missing from the manuscript I reviewed.

The authors did not specify how many times students were contacted to participate in the survey, particularly those who did not respond to the first (only?) invitation. Given that a little over half of those invited (55.1%) responded, subsequent invitations to non-respondents may have increased the response rate.

It was interesting to note that women constituted some 75% of respondents. Is this representative of the percentages of women in German medical education as a whole?

Some 10 medical schools had no respondents. Was there any reason for this?

It is most reasonable that students would "feel less informed about how it (medical education) would proceed," given that most faculties are themselves struggling this this and many attendant issues. The current state of such decision-making would be a useful addition to the discussion, but is not mandatory.

Recommendation: publish as presented.

Reviewer #2: The authors present an interesting study investigating COVID-19-related stressors and expectations of medical students towards education during the pandemic in a German cohort of 679 medical students.

The authors conclude that the participating students were aware of the pandemic and its consequences regarding both in the medical and academic context.

However, there are several methodological issues, which need to be addressed.

Introduction:

- All in all, a clear introduction with a comprehensible aim of investigation.

Methods:

- Besides the validated parts of your survey (STAI, GAD-7, PHQ-9): how was the questionnaire validated? Is it based on the literature? Was it pilot tested?

- As this is an online-based, cross-sectional study, did you use the STROBE or CHERRIES checklist?

- For better understanding, please consider providing your questionnaire as supplementary material.

Results:

- P8, I160,161: As 372 participants fully completed the questionnaire, how did you deal with the non-responders? If you separately included them in your analysis as well, please consider providing the particular completion rate for each question of the questionnaire.

- As you chose for a 7-point Likert scale, why did you break down your results on mean and SD? Please consider providing more detailed results for each particular question.

- Table 1 is noted in the text, but missing in the data.

Discussion & Conclusion:

- P13, I297-299: “It can be hoped that the current wave of digitalisation will not recede but push medical education into a real digital transformation“. Very interesting aspect! However, as some participants of your survey are „worried about the disadvantages of studying online (...)“ (P10, I222), you should consider also taking these critical statements into account. Furthermore, are there maybe parts of medical education being (or at least seeming) unsuitable for digitization?

- P13, I303, 304 (limitations): Regarding your sample size and study set-up, please consider discussing various types of bias and how you dealt with them - e.g. selection bias due to the way you recruited your participants, selection bias due to the fact that interested students are more likely to complete the questionnaire, or social desirability, meaning that participants choose the answer that they assume is favorable. Which methods against bias did you employ?

Language:

- Overall, the manuscript is well written and comprehensible.

6. PLOS authors have the option to publish the peer review history of their article (what does this mean?). If published, this will include your full peer review and any attached files.

Reviewer #1: No

Reviewer #2: **Yes: **Roman Kloeckner

---

## [Author Response · Author response to Decision Letter 0]

16 Oct 2020

Dear Prof. Heber, 

Thank you very much for consideration of the revised version of our manuscript “Medical education in times of Covid-19: German students’ expectations – a cross-sectional study”.

We highly appreciate the effort reviewers have to undergo and are very grateful for their contributions to the scientific community in general and our manuscript in particular. 

Please find below our detailed responses to the comments of reviewer 1 and reviewer 2. 

Reviewers' comments:

Reviewer #1:

Note: Table 1 was missing from the manuscript I reviewed.

Thank you for this hint, we apologise for this misunderstanding and have uploaded Table 1. 

The authors did not specify how many times students were contacted to participate in the survey, particularly those who did not respond to the first (only?) invitation. Given that a little over half of those invited (55.1%) responded, subsequent invitations to non-respondents may have increased the response rate.

Thank you for this valid point. Unfortunately, it was only one contact but we absolutely agree that in future studies subsequent invitations will be sent to participants to increase response rate. 

It was interesting to note that women constituted some 75% of respondents. Is this representative of the percentages of women in German medical education as a whole?

Thank you for your comment. The 75% are not representative of the percentages of women in German medical education. Figures from winter term 18/19 show 62% female medical students. We added a corresponding sentence to the discussion. 

Some 10 medical schools had no respondents. Was there any reason for this?

This is a good question. It was an open invitation with no control over response behaviour. The ap-proach to each medical school was always the same. We can’t give the answer why some places had no participants. Analyses for patterns also showed no results. To clarify, we have added a corresponding sentence in the discussion.

It is most reasonable that students would "feel less informed about how it (medical education) would proceed," given that most faculties are themselves struggling this this and many attendant issues. The current state of such decision-making would be a useful addition to the discussion, but is not mandatory.

Thank you for your valuable comment. We added a sentence to the discussion

Recommendation: publish as presented.

Reviewer #2: The authors present an interesting study investigating COVID-19-related stressors and expectations of medical students towards education during the pandemic in a German cohort of 679 medical students.

The authors conclude that the participating students were aware of the pandemic and its consequences regarding both in the medical and academic context.

However, there are several methodological issues, which need to be addressed.

Introduction:

- All in all, a clear introduction with a comprehensible aim of investigation.

Thank you for your appreciation. 

Methods:

- Besides the validated parts of your survey (STAI, GAD-7, PHQ-9): how was the questionnaire validated? Is it based on the literature? Was it pilot tested?

Thank you for your comment. The construction of the questionnaire was based on existing literature. The items were carefully chosen including expert discussions; however, there was no pilot test as such. We added a clarifying sentence in the methods. 

- As this is an online-based, cross-sectional study, did you use the STROBE or CHERRIES checklist?

CHERRIES

We mainly used the STROBE checklist as we focused on the cross-sectional study design and also had no classical online study as such. We still also considered the CHERRIES checklist where applicable. Additionally, we used a reliable software programme checked by our University (questback) for the assessment rendering many points of the CHERRIES checklist no longer relevant. 

- For better understanding, please consider providing your questionnaire as supplementary material.

In general we are not opposed to providing the questionnaire. However, as it was a German study, the questionnaire is also in German. To be scientifically sound, we would need to do a forward backward translational process with certified translators. As big parts of the questionnaire are official ones (STAI, GAD-7, PHQ-9) anyway, we thought that a mere description in the body of the manuscript was sufficient for the message of the paper. Thus, we restrained from a translation process. However, if this is a requirement for publication, we are more than happy to do so. Please let us know.

Results:

- P8, I160,161: As 372 participants fully completed the questionnaire, how did you deal with the non-responders? If you separately included them in your analysis as well, please consider providing the particular completion rate for each question of the questionnaire.

Thank you for your comment. We excluded non-responders and included all questionnaires that were filled in at least 80%. We added a sentence in the method section. 

- As you chose for a 7-point Likert scale, why did you break down your results on mean and SD? Please consider providing more detailed results for each particular question.

As the data of the 7 point Likert scale were normally distributed we decided to present mean and standard deviations as relevant significant values (see also Fahrmeir et al., 2016). Further, we did not provide more detailed results as this would lead to over-presentation of data. As there was no intervention in this study, we did not calculate any effect sizes. Of course, we are happy to present further statistical values. Just let us know which ones you consider relevant. 

- Table 1 is noted in the text, but missing in the data.

Thank you for this hint, we apologise for this misunderstanding and we uploaded Table 1.

Discussion & Conclusion:

- P13, I297-299: “It can be hoped that the current wave of digitalisation will not recede but push medical education into a real digital transformation“. Very interesting aspect! However, as some participants of your survey are „worried about the disadvantages of studying online (...)“ (P10, I222), you should consider also taking these critical statements into account. 

Thank you for your comment. We added this point to the discussion. 

Furthermore, are there maybe parts of medical education being (or at least seeming) unsuitable for digitization? 

Thank you for your comment. We added this point to the discussion. 

- P13, I303, 304 (limitations): Regarding your sample size and study set-up, please consider discussing various types of bias and how you dealt with them - e.g. selection bias due to the way you recruited your participants, selection bias due to the fact that interested students are more likely to complete the questionnaire, or social desirability, meaning that participants choose the answer that they assume is favorable. Which methods against bias did you employ?

Thank you, we are aware of the selection bias that students who are interested in this topic are more willing to complete this questionnaire. We tried to reduce this risk by contacting all German medical students via Email through the German national student body. However, as the study was on voluntary base we could not completely exclude this risk. We looked for patterns in the participants but could not find any. Further, as the survey was anonymous and the participants had no contact to the investigator we think that the percentages of students acting socially desirable might be very low. However, it is a valid point to talk about biases and so we have added these points to the discussion. 

Language:

- Overall, the manuscript is well written and comprehensible.

Thank you for this appreciative comment. 

We hope that we could satisfyingly address all issues and concerns and that our manuscript in its revised version is now suitable for publication.

If you have any further questions, please do not hesitate to contact us any time

Thank you for your consideration, and we look forward to hearing from you.

Sincerely,

Anne Herrmann-Werner

---

## [Editor Report · Decision Letter 1]

20 Oct 2020

Medical education in times of Covid-19: German students’ expectations

– a cross-sectional study

PONE-D-20-21365R1

Dear Dr. Herrmann-Werner,

We’re pleased to inform you that your manuscript has been judged scientifically suitable for publication and will be formally accepted for publication once it meets all outstanding technical requirements.

Kind regards,

Jenny Wilkinson, PhD

Academic Editor

PLOS ONE

Additional Editor Comments (optional):

Thank you for your responses and revisions; these have satisfactorily addressed the reviewer comments of the previous version.
---

## [Editor Report · Acceptance letter]

10 Nov 2020

PONE-D-20-21365R1 

Medical education in times of Covid-19: German students’ expectations
– a cross-sectional study 

Dear Dr. Herrmann-Werner:

I'm pleased to inform you that your manuscript has been deemed suitable for publication in PLOS ONE. Congratulations! Your manuscript is now with our production department. 

Kind regards, 

on behalf of

Dr Jenny Wilkinson 

Academic Editor

PLOS ONE